# Delivery system can vary ventilatory parameters across multiple patients from a single source of mechanical ventilation

Kyle K. VanKoevering[1,2]*, Pratyusha Yalamanchi[1], Catherine T. Haring[1], Anne G. Phillips[3], Stephen Lewis Harvey[3], Alvaro Rojas-Pena[3], David A. Zopf[1], Glenn E. Green[1]

1 Department of Otolaryngology, Head & Neck Surgery, Michigan Medicine, University of Michigan, Ann Arbor, MI, United States of America, 2 Department of Otolaryngology, Head and Neck Surgery, Wexner Medical Center, Ohio State University, Columbus, OH, United States of America, 3 Department of Surgery, Section of Transplantation, Michigan Medicine, University of Michigan, Ann Arbor, MI, United States of America

* kylevk@umich.edu

## Abstract

### Background

Current limitations in the supply of ventilators during the Covid19 pandemic have limited respiratory support for patients with respiratory failure. Split ventilation allows a single ventilator to be used for more than one patient but is not practicable due to requirements for matched patient settings, risks of cross-contamination, harmful interference between patients and the inability to individualize ventilator support parameters. We hypothesized that a system could be developed to circumvent these limitations.

### Methods and findings

A novel delivery system was developed to allow individualized peak inspiratory pressure settings and PEEP using a pressure regulatory valve, developed de novo, and an inline PEEP 'booster'. One-way valves, filters, monitoring ports and wye splitters were assembled in-line to complete the system and achieve the design targets. This system was then tested to see if previously described limitations could be addressed. The system was investigated in mechanical and animal trials (ultimately with a pig and sheep concurrently ventilated from the same ventilator). The system demonstrated the ability to provide ventilation across clinically relevant scenarios including circuit occlusion, unmatched physiology, and a surgical procedure, while allowing significantly different pressures to be safely delivered to each animal for individualized support.

### Conclusions

In settings of limited ventilator availability, systems can be developed to allow increased delivery of ventilator support to patients. This enables more rapid deployment of ventilator capacity under constraints of time, space and financial cost. These systems can be smaller,

**Data Availability Statement:** All relevant data are within the manuscript and its Supporting Information files.

**Funding:** This study was supported by a private donation to the University of Michigan (KVK). The funders had no role in study design, data collection and analysis, decision to publish, or preparation of the manuscript.

**Competing interests:** I have read the journal's policy and the authors of this manuscript have the following competing interests: KVK, GEG and DAZ are founders of MakeMedical LLC, which is the legal manufacturer of this device (provided at cost). This does not alter our adherence to PLOS ONE policies on sharing data and materials.

lighter, more readily stored and more rapidly deployable than ventilators. However, optimizing ventilator support for patients with individualized ventilation parameters will still be dependent upon ease of use and the availability of medical personnel.

## Introduction

Modern ventilators are complex mechanical and electronic devices that enable a patient to receive life sustaining respiratory support. The worldwide COVID-19 pandemic has highlighted an inadequate reserve capacity of ventilators for the number of individuals needing mechanical intervention during a mass disaster [1]. While it might seem straightforward to increase the supply of ventilators during pandemics, doing so is problematic due to the amount of time required to develop production systems, the risk of mechanical or electronic failure in ventilators being brought online, difficulties in upgrading dilapidated stored ventilators and training requirements for individuals to use new ventilators. Moreover, in space-constrained environments, in the developing world, and during environmental disasters there may be sharp limitations in the ability to store, maintain and even deploy ventilators that may only rarely see use. We thus expect that during current and future disasters, conditions will continue to exist where multiple individuals simultaneously require ventilator support while mechanical ventilator supply is inadequate.

The traditional answer to this has been two-fold: triage, namely intentionally not supporting those who could potentially benefit from treatment; or rationing, namely limiting the use of ventilators based upon varying factors, some of which may be ethically questionable. Doctors around the world are already making challenging choices as to prioritization of limited supply of ventilator support given excess patient demand [2]. While the private sector has committed to producing new ventilators, production and distribution take time that our healthcare providers and critically ill patients simply do not have. Therefore, the Ventilator Emergency Use Authorization (EUA) was issued by the United States Food and Drug Administration in response to concerns relating to insufficient supply. Similar authorization tracks have been developed in some, but not all, countries.

In response to this ventilator resource crisis, attempts have been made to increase ventilator capacity through so called "vent splitting." Until recently, all methods of connecting multiple patients to ventilators required that patients sharing a ventilator be approximately the same size, have the same level of lung injury, and tolerate the same ventilator settings [3]. These systems result in cross contamination, are difficult to use, consistently failed optimization for the individual COVID-19 patient requirements, and were challenging to execute in times of need. The use of non-individualized ventilator splitting was specifically condemned in a statement from several international medical organizations due to these concerns [4].

We hypothesized that a system could be developed to allow individualized ventilator parameters to be used across patients from a single source of mechanical ventilation to enable the capability of delivering ventilation to those who would otherwise not receive it. We hypothesized that a system could further adapt to relevant clinical scenarios including varying patient physiologies, surgical insult, coughing, disconnection and movement.

## Materials and methods

We sought to develop a straightforward, streamlined system *de novo* to allow a single ventilator to support a second patient with individualized pressure control settings. Several rounds of prototyping were initially performed with 3D-printed constructs to enable optimization of

components. The core of the optimal design identified (VentMI) utilizes an inspiratory pressure regulator, an inline booster for the positive end expiratory pressure [PEEP booster] and conventional Wye-piece splitting system to expand a single ventilator's airflow into separate circuits. The pressure regulator is connected in-line with the inspiratory circuit to allow individualized inspiratory pressure control. Incremental PEEP control is provided by the PEEP booster. To complete a medically usable system, the pressure regulators were paired with one-way flow valves to ensure pressures do not equilibrate across the circuits and viral/bacterial filters to limit risks of cross-contamination. The entire system is preassembled for immediate operational use, limiting potential confusion and set-up time.

The final system is manufactured and assembled in an ISO 13485 medical facility (Autocam Medical, Grand Rapids, MI, USA) under clean conditions. Medical grade adhesive is used to secure connectors and valves to the wye pieces, and one side of the splitter was capped to allow rapid deployment in stand-by mode [Fig 1A]. Quality control testing is performed on each manufactured regulator and PEEP booster to ensure intended performance. Internal components are machined from medical-grade Radel® (polyphenylsulfone) and the housing is machined from aluminum, with a silicone (platinum-cured polydimethyl siloxane) gasket seal and stainless-steel spring. Each regulator is sanitized with ethanol sonication prior to final assembly.

Initial deployment in "stand-by mode" is to a stable patient on a ventilator. At any later point, another patient can then be rapidly connected to the attachment sites for the system [Fig 1B] to use the same ventilator.

### Inspiratory pressure regulator

The final inspiratory pressure regulator was designed from the concepts of a commercial SCUBA regulator system, with components specifically designed to function across the physiologic range of expected ventilation pressures. A two-chamber system was utilized where the upper chamber is sealed from the lower chamber by a moving piston when the target pressure is reached. Once sealed, airflow into the upper chamber is halted and pressure in the inspiratory limb remains stable. The pressure at which the chambers seal can be variably adjusted by modifying the spring compression via a screw adjusted cap. The initial prototypes for the regulator were manufactured with 3D printing (Form2, FormLabs, Somerville, MA, USA), and subsequently transitioned to a machined medical-grade aluminum product manufactured in a GMP, ISO compliant medical machining facility (AutoCam Medical). This system was found to offer substantive advantages over both volume or flow limited systems. The device contains a silicone gasket, currently fabricated out of biocompatible medical-grade, platinum-cured polydimethyl siloxane.

### PEEP booster

To regulate PEEP, a previously developed inline ball valve was manufactured from the original schematics (Boehringer Laboratories, PEEP Valve Kit, Phoenixville, PA, with permission). These PEEP boosters are placed in-line on the circuit and were only removed from widespread use as mechanical ventilators were developed that could provide these settings internally rather than an external device. The PEEP Booster is a ball-valve system that utilizes the weight of a 5/8 inch ball in a tapered chamber to provide a constant pressure gradient across variable flow. Balls of various specific gravity (Nylon, Teflon, Stainless Steel) allow for a 2, 4, and 8cm H2O pressure gradient on testing.

### Pressure monitoring

Development of a simple, widely accessible patient-specific monitoring system is critical for the success of this circuit. We utilized a conventional arterial line pressure transducer connected to a

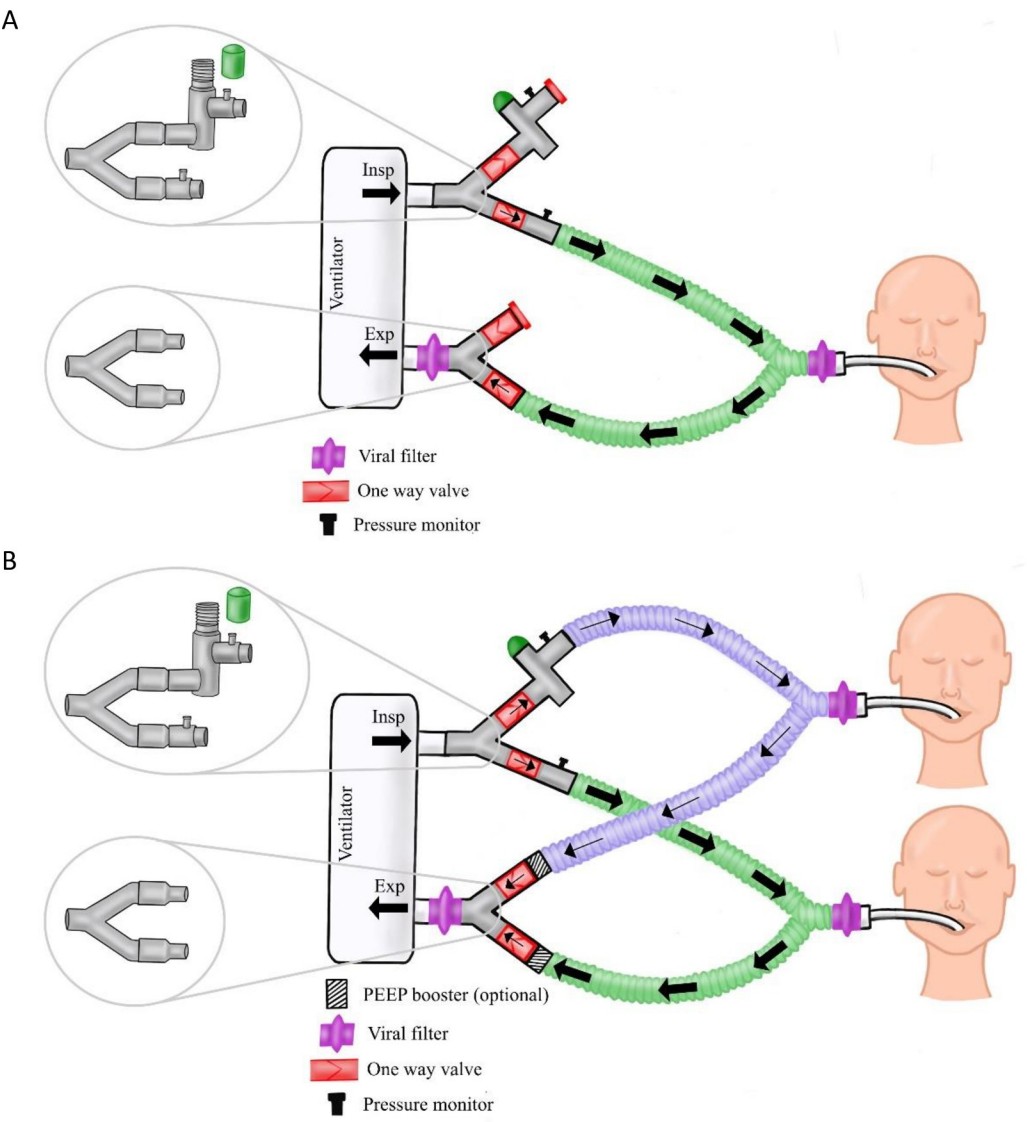

**Fig 1.** Schematic diagram showing how a single patient could be connected to the VentMi system in 'stand by' mode, awaiting a second patient (A) and schematic of both patients connected to VentMi with one patient having inspiratory pressure downregulated by the regulator, and the second controlled by the ventilator.

conventional vital monitor to follow the ventilatory pressure of each patient in real-time. The pressure loops are displayed on the patient's vital monitor, and a simple conversion from mmHg to cmH20 can be performed by multiplying the mmHg value by 1.36. The transducer can be connected (dry) to a standard luer lock port anywhere in the circuit between the patient and the regulators and provides real-time feedback on the individual pressure loops for each patient.

## System assembly

Inspiratory and expiratory components were identified to complete the system. Branched adapters attach directly to the circuit. A system was developed to allow for partial predeployment on a stable patient (see Fig 1). In this configuration, the first patient can be maintained indefinitely without clinically meaningful changes to flow. We attached intravenous tubing to

side ports of the adapters to allow for pressure monitoring through pressure transducers. These ports are part of the assembly and can be used clinically. The wye splitters are connected to directionally specific one-way valves and the desired pressure regulator system (inspiratory pressure regulator or PEEP boosting system respectively) and are pre-assembled as separately packaged inspiratory and expiratory units. Preassembling the units with one-way valves decreases the assembly time for deployment and reduces risk of incorrect assembly.

### *In-Vitro* testing

Preliminary testing was first performed on linear lung simulation balloons. Leak testing of the circuit was performed, with the VentMi system connected to an anesthesia gas machine (GE Datex-Ohmeda Aisys Carestation, General Electric, Boston, MA, USA). A leak test was first performed using an inspiratory pressure of 50 cm H20 and PEEP of 10, with Rate of 10/min and I:E ratio of 1:2 with flow of 15 L/m. Next, a full inspiratory hold at 60 cm H20 was performed. Various pressure control settings were tested to validate performance. Next, a robust testing sequence was subsequently performed on a Puritan Bennet 840 ICU Ventilator [PB840] (Medtronic, Minneapolis, Minnesota, USA). Pressure control mode was used with a 1L test balloon on circuit 1 and a 3L test balloon on circuit 2 to test unmatched patients. A full range of physiologic pressures were tested.

### Cycle testing

Repetitive cycle testing was then performed such that a regulator was connected to a modified ventilator circuit with rapid cycling to test durability of the regulator. The system was set for the regulator to downregulate pressure to 12 cmH20 while the ventilator drove inspiratory pressures at 25 cmH20. The regulator was then rapid-cycled at 96 breaths per minute.

### *In-Vivo* testing

We further investigated this model in animal tests using porcine and ovine models, to determine whether independent, lung protective ventilation could be delivered to two patients connected to one ventilator. All animal studies were carried out in strict compliance with the Guide for the Care and Use of Laboratory Animals of the National Institutes of Health. The protocol was reviewed by the University of Michigan University Committee on Use and Care of Animals (UCUCA) for the single pig feasibility study, and the Charles River Animal Use Committee for the combined pig and sheep study. Both studies were approved by the respective animal use committees. At the conclusion of each study, animals were sacrificed with a combination of sodium pentobarbital and bilateral pneumothoraxes.

**Single animal test.** A standard porcine model was chosen for the single animal test given respiratory physiology similar to human physiology and prior use modeling respiratory changes. Specifically, a healthy female swine weighing 71 kg, was sedated with intramuscular mix of 5 mg/kg Tiletamine HCl and Zolepam HCl and 3 mg/kg Xylazine and subsequently intubated with an appropriately sized cuffed endotracheal tube and mechanically ventilated (MV) with 47% FiO2. Total IV anesthesia (TIVA) was maintained with a propofol infusion. Ventilator settings were adjusted to maintain peak inspiratory pressures <20 cmH2O to the swine and CO2 target between 35–45 mmHg. A catheter was placed via the internal jugular vein for administration of fluids and monitoring of central venous pressure. The VentMi circuit was connected to the porcine model with a 3L linear lung simulator on the other circuit and a limited volume ventilator. A series of clinical stressors and functional tests were performed with the VentMI system to evaluate the safety of the device. At the end of each intervention [Table 2], a 15 minute acclimation period was allowed for the animal prior to data collection.

Arterial blood gas (ABG) values were compared during the protocol to ensure adequate ventilation was maintained. The porcine model was connected to a balloon lung simulator via the splitting mechanism proposed.

**Pig-sheep dual animal testing.** In order to validate the performance of the VentMi system, a dual large animal model was tested with two animals of different physiology: a 43.5 kg female Dorsett crossbred sheep and an 86 kg male hybrid Yorkshire pig. Both animals were sedated, intubated and initially ventilated via separate veterinary anesthesia ventilators following the above protocol. Total intravenous anesthesia was administered for both animals without paralysis. A DigiVent DVX8 large animal portable ventilator (Digicare Biomedical Technology, Boynton Beach, FL, USA) was selected as the primary ventilator for the split ventilation, due to its capacity for pressure control ventilation. The swine was initially placed on the Digivent ventilator in standard pressure control ventilation in "stand-by" mode [Fig 1A] with VentMi connected but the regulated circuit capped, allowing normal ventilation to just one patient. The sheep was then connected to the pig's ventilator concurrently via the VentMi system [Fig 2]. Minimal adjustments were made to the ventilator to accommodate the increased flow and minute ventilation. The animals were then subjected to a variety of physiologic scenarios, including 1) matched ventilation; 2) individualized ventilation (4cm PEEP boost and increased PIP to the swine, baseline pressures to the sheep via inspiratory regulator); 3) circuit occlusion; 4) physiologic stressor to one animal (superficial flap surgery performed on the swine); and 5) maximal pressure differential (increased PIP and PEEP of swine until cardiac instability was noted, while maintaining stable pressures in the sheep). Data was collected automatically through SurgiVet Data Logger System (Smith's Medical, Minneapolis, MN, USA) at 30 second intervals.

## Results

Initial prototyping revealed that a flow restriction technique was not as reliable or safe as a pressure controlled system, as circuit occlusion on one side could result in significant barotrauma to the second patient under volume control. Further prototyping confirmed a true pressure regulator was necessary for reliable performance. Large volume leaks (i.e. popoff valves or pressure relief valves) result in ventilator alarms, inadequate flow in certain ventilators, and concerns regarding aerosolized viral particles. One-way valves were imperative to maintain any desired pressure differential between circuits and prevent pressure equilibration, and serve a secondary function of limiting potential cross contamination. Placement of one-way valves in reverse orientation was a potential problem, and prompted the pre-assembly of a fully functional system. By sealing the system, disconnects and incorrect placement could be avoided. Placement of labels and flow direction arrows was felt to simplify the system.

Weight of the combined system was 1.2 kg and size was 27 cm x 23 cm x 9 cm. Since each circuit is connected to the endotracheal tube in a similar fashion to standard ventilation, a closed-line suction system can be sued in each patient to limit aerosolization.

### System deployment

With the preassembled VentMi system, deployment of the system required less than 1 minute to connect the system in stand-by mode for a respiratory therapist unfamiliar after a training lasing 8 minutes with the device, and less than 30 seconds to add the second patient on simulated testing.

In dual animal testing, placement of large animals into stand-by mode or dual ventilation mode took 25 seconds and 12 seconds respectively.

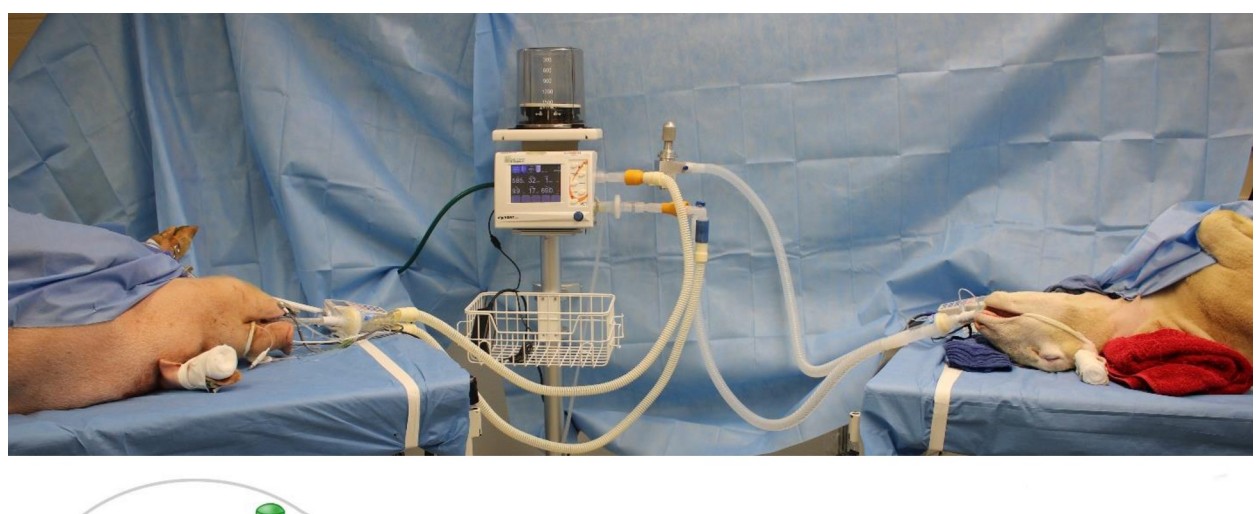

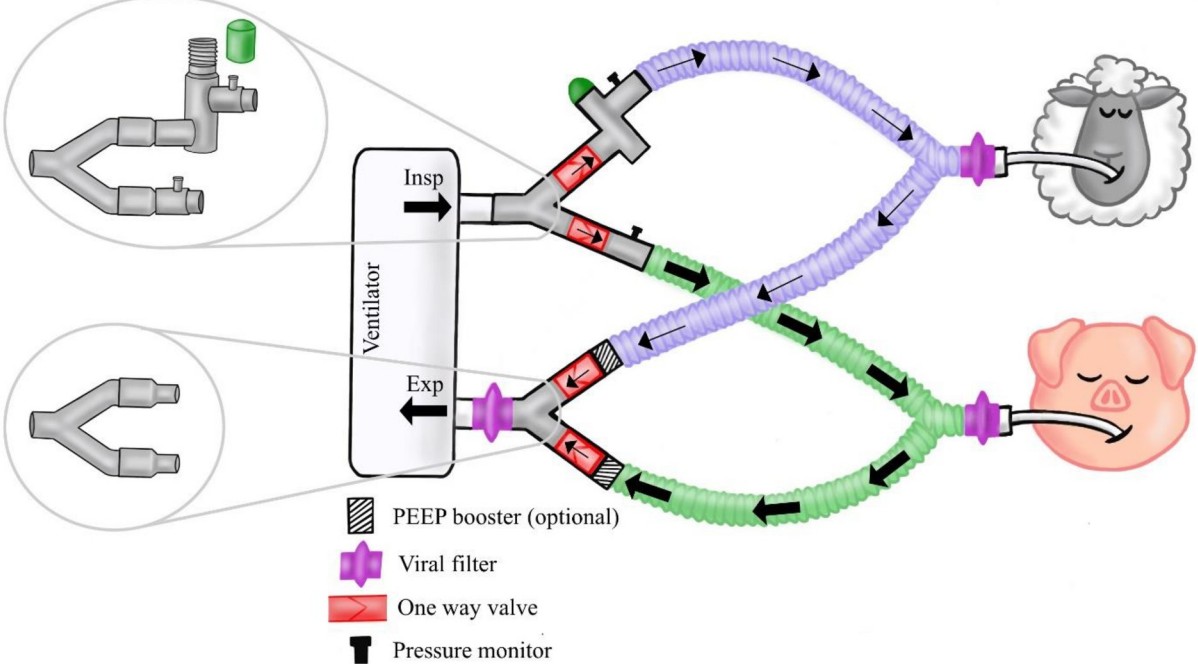

**Fig 2. Picture and schematic diagram of the connection of the dual animal study to the VentMi circuit.** The pig's PIP was controlled by the ventilator and PEEP upregulated by PEEP boosters during the 6 hour co-ventilation experiment, while the sheep was regulated by the inspiratory regulator with the ventilator controlling the PEEP.

### *In-Vitro* testing

Leak testing on the anesthesia gas machine demonstrated a negligible leak of less than 100 mL for all breaths at prescribed ventilation parameters, with no visible change in the plateau pressure over 2 seconds of inspiratory hold.

Initial performance testing was completed with identical simulation lungs attached to each circuit, with VentMi connected to the anesthesia machine. The machine was set to pressure control ventilation a peak inspiratory pressure of 36 cmH$_2$0 while the pressure regulator was variably dialed to a range from 12 cmH$_2$0 up to 36 cmH$_2$0. The PEEP was initially set to 5 cm H20 on the anesthesia machine, and PEEP boosters were then added to the circuit, confirming a fully individualized pressure control could be maintained on each circuit. For any desired

**Table 1. Pressure results of benchtop testing on PB840 ICU ventilator with VentMi system.**

| Vent Settings: | Frequency: 10 per minute | | I2E: 1:2 | P Slope 50% | Pressure Control | | A/C |
|---|---|---|---|---|---|---|---|
| | | Balloon 1 (PIP/PEEP) | Balloon 2 (PIP/PEEP) | | | | |
| **Inspiratory Pressure Regulator Testing** | | | | | | | |
| PIP (cmH2O) | PEEP (cmH2O) | Unregulated (mmHg) | 90% PIP (mmHg) | 80% PIP (mmHg) | 60% PIP (mmHg) | Vt Unregulated (mL) | Vt 60% PIP Balloon 2 (mL) |
| 15 | 5 | 11/4 | 10/4 | 9/4 | 7/4 | 375 | 222 |
| 20 | 5 | 14/4 | 12/6 | 11/4 | 8/4 | 612 | 253 |
| 25 | 5 | 18/4 | 16/4 | 14/4 | 11/4 | 834 | 425 |
| 30 | 5 | 22/4 | 20/4 | 17/4 | 13/4 | 1113 | 607 |
| 20 | 10 | 15/8 | 13/8 | 12/8 | 9/8 | 375 | 110 |
| 25 | 10 | 18/8 | 16/8 | 14/8 | 11/8 | 615 | 271 |
| 30 | 10 | 22/8 | 20/8 | 18/8 | 13/8 | 873 | 373 |
| 35 | 10 | 26/8 | 23/8 | 21/8 | 16/8 | 1223 | 603 |
| 25 | 15 | 18/12 | 16/12 | 14/12 | 12/12 | 403 | 84 |
| 30 | 15 | 22/12 | 20/12 | 18/12 | 13/12 | 672 | 177 |
| 35 | 15 | 26/12 | 23/12 | 21/12 | 16/12 | 975 | 351 |
| 40 | 15 | 29/12 | 26/12 | 23/12 | 17/12 | 1387 | 529 |
| 30 | 20 | 22/15 | 20/15 | 17/15 | 15/15 | 510 | 90 |
| 35 | 20 | 26/15 | 23/15 | 21/15 | 16/15 | 905 | 156 |
| 40 | 20 | 29/15 | 26/15 | 23/15 | 17/15 | 1332 | 285 |
| 45 | 20 | 33/15 | 30/15 | 27/15 | 20/15 | 1811 | 615 |
| **PEEP Booster Testing** | | | | | | | |
| 20 | 5 | 14/4 | | | 11/4 | | |
| 20 | 5 | 14/6 | | | 11/4 | | + 2 PEEP Balloon 1 |
| 25 | 10 | 18/10 | | | 14/8 | | + 2 PEEP Balloon 1 |
| 30 | 15 | 22/14 | | | 17/12 | | + 2 PEEP Balloon 1 |
| 25 | 10 | 18/11 | | | 14/8 | | + 4 PEEP Balloon 1 |
| 25 | 10 | 18/12 | | | 14/8 | | + 2 + 4 PEEP Balloon 1 |
| 25 | 10 | 18/14 | | | 14/8 | | + 8 PEEP Balloon 1 |

ventilation pressures, the ventilator was set to the highest planned Peak Inspiratory Pressure and the lowest planned PEEP.

Similarly, a robust benchtop testing performed on the PB840 tested ventilation pressures ranging from 15/5 to 45/20 (PIP/PEEP), confirming the performance of the VentMI system. Regulated pressures were tested from 60% to 100% of the PIP at all testing conditions with success, and the 2, 4, and 8 cmH20 PEEP boosters all functioned as intended, allowing a boost in PEEP from 2–14 cm H20 above baseline (boosters can be "stacked" as needed). Various I:E ratios were tested ranging from 1:1–1:5 as well as pressure slope from 5–100%, all of which demonstrated reliable performance in the regulator across the full spectrum of pressure control ventilation [Table 1].

## Cycle testing

The rapid-cycle testing completed 600,000 continuous cycles on one of the VentMi regulators over 5 days at 96 cycles/minute. Ventilator pressure remained stable throughout the test at 25 cm H20 and the regulator maintained a downregulated pressure of 12 cmH20 stably throughout the test. The regulator was subsequently disassembled, with no evidence of wear or degradation on the system.

**Table 2. Testing conditions performed with single-animal swine model.**

| Testing Condition | Tidal Volume (mL) | Respiratory Rate (B/min) | FiO2 (%) | PEEP | End Tidal CO2 (mmHg, Pig) | Circuit 1: Balloon PIP (Mean cmH20) | Circuit 1: Balloon PEEP (Mean cmH20) | Circuit 2: Swine PIP (Mean cmH20) | Circuit 2: Swine PEEP (Mean cmH20) | Comments |
|---|---|---|---|---|---|---|---|---|---|---|
| **Stand-by Ventilation of Pig** | 950 | 10 | 49 | 3 | 38 | N/A | N/A | 15.1 | 4.1 | Circuit 2 capped |
| **Balloon + Pig baseline** | 1140 | 10 | 48 | 3 | 39 | 15.2 | 5 | 12.7 | 4.3 | Airway 1: Balloon / Airway 2: Pig |
| **Balloon + Pig Regulated** | 1610 | 10 | 47 | 3 | 39 | 19.3 | 5.3 | 14.8 | 4.8 | Regulated PIP to 14 in Pig |
| **Coughing** | NA | 8 | 48 | 3 | Unable to evaluate | 38.2 | N/A | 8.2 | N/A | Simulated coughing (squeezing balloon), minimal changes circuit 2 |
| **Dyssynchrony** | N/A | 44-Circuit 1 8-Circuit 2 | 48 | 3 | 45 | N/A | N/A | 11.1 | 4.6 | Over-breathing (44 B/Min) in Circuit 1, no significant change in pressures for Circuit 2 |
| **Open Circuit** | N/A |  | 48 | N/A | 54 | 1.2 | 1 | 3.40 | 1 | Disconnected circuit 1 resulted hypoventilation for Circuit 2 |
| **In Line PEEP regulator** | 1260 | 12 | 49 | 3 | 67 | 17.8 | 3.9 | 14.3 | 6.8 | 2 cm PEEP booster on Swine provides reliable PEEP increase but results in hypoventilation |

### *In-Vivo* testing

**Single animal test.** The study was conducted over a five-hour total duration. The pig was euthanized at the completion of testing per protocol. The results demonstrated that the pig could be safely ventilated at stable pressures while varying the pressures delivered to the balloon across a wide range. The study also confirmed that a standard arterial line pressure transducer provides excellent real-time monitoring of ventilation pressures, and although the ventilator was limited to volume-controlled ventilation modes, it was run based off pressure readings analogous to pressure control ventilation. Simulated coughing and dyssynchrony in the balloon (manual squeezing) did not result in significant ventilatory changes to the swine due to the function of the 1-way valves [Fig 3]. In open and occluded circuit scenarios, the swine was protected from barotrauma, but did demonstrate signs of hypoventilation that were immediately alarmed on the ventilator and monitors. The study demonstrated the pressure regulator could safely control the inspiratory pressure to the swine from 12–17 cm H20 while the inspiratory pressure in the balloon was increased up to 22 cm H20, while maintaining stable ventilation for the swine [Table 2]. Arterial blood gas monitoring confirmed the pig remained well ventilated with stable parameters even when inspiratory pressures were regulated by the pressure regulator. The PEEP boosters provided reliable increases in PEEP on the applied circuit. Given that the system is entirely housed at the ventilator, movement of the swine and balloon resulted in no changes to the performance.

**Disparate dual animal test.** The dual animal study was completed over a 6 hour duration. The small DigiVent ventilator is a portable ventilator that was readily capable of generating adequate volume and flow to support two large animals from the single ventilator, and was readily assembled as an ICU style ventilator. There were no complications during the testing. Connecting the swine into stand-by mode required 25 seconds, while connecting the sheep to the second circuit required 12 seconds. Small adjustments were made to the ventilator settings

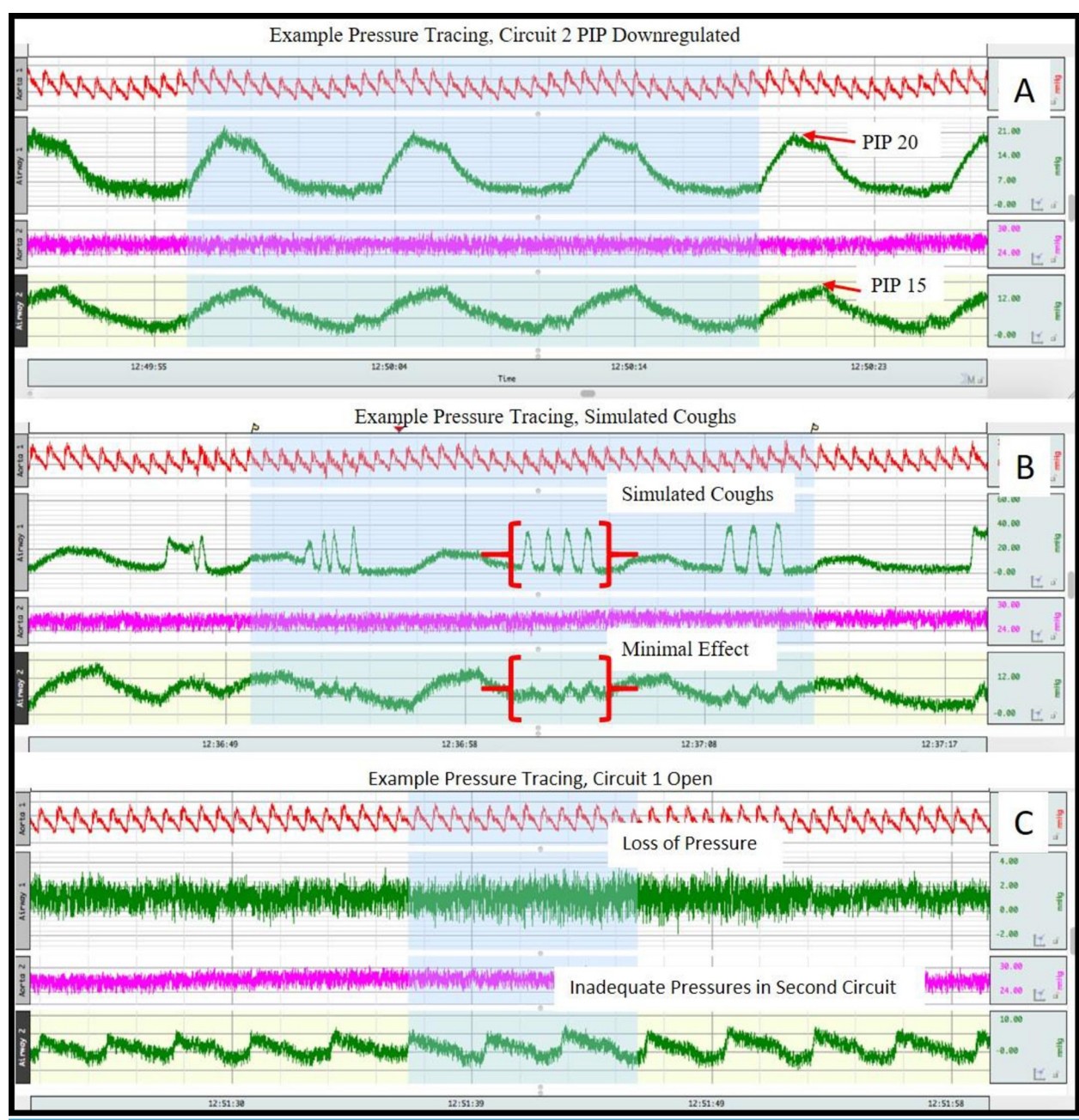

**Fig 3. Pressure tracings from single-patient swine model demonstrating the balloon [top green tracing] compared to the pig [bottom green tracing] ventilation profiles.** (A) demonstrates the downregulated PIP for the pig. (B) demonstrates the minimal effect on the pig from simulated coughing [squeezing] in the balloon. (C) demonstrates the global loss of pressure seen with an open circuit [disconnected balloon].

after split ventilation was established to accommodate for the increased flow and volume requirements of the ventilator (increased respiratory rate by 1 BPM). We tested several scenarios for at least 15 minutes duration, including matched ventilation, various increased pressures for the swine, and a superficial surgical flap dissection in the swine to simulate physiologic stress. The flap procedure consisted of elevating and exposing the abdominal skin and soft tissues with electrocautery, and lasted 40 minutes. During the procedure, the animals remained very stable with no need for adjustments. Comparison of the ventilatory data demonstrates

Table 3. Comparison of swine and sheep ventilatory parameters at various testing conditions.

| Testing Condition | Ventilatory Parameters for Each Testing Condition | | | | Measured Ventilator Findings for Swine Subject | | | Measured Ventilator Findings for Sheep Subject | | | Comments |
|---|---|---|---|---|---|---|---|---|---|---|---|
| | RR (breaths/min) | FiO$_2$ (%) | Vent Setting ΔP (cmH$_2$0) | Vent Setting PEEP (cmH$_2$0) | Mean PIP/PEEP Measured (Swine) | End Tidal CO2 (Swine) | SpO2 (%) (Swine) | Mean PIP/PEEP Measured (Sheep) | End Tidal CO2 (Sheep) | SpO2 (%) (Sheep) | |
| **Individual Ventilator** | 16 | 62 | NA | NA | 19.2/3.8 | 40.8 | 96.1 | 19.7/2.7 | 45.8 | 96.8 | Each animal ventilated on separate ventilator |
| **Initial Pairing** | 17 | 66 | 16 | 5 | 21.2/4.96 | 44.4 | 97.1 | 21.4/4.0 | 48.9 | 96.5 | Initially paired without individualization, nearly identical profiles |
| **4cm PEEP (swine)** | 17 | 65 | 22 | 2 | 23.7/6.9 | 42.6 | 98.3 | 19.5/1.33 | 47.3 | 96.2 | Intentionally ventilated with individualized pressures including 4cm PEEP booster for swine |
| **Sheep Occluded** | 17 | 67 | 22 | 2 | 22.2/6.6 | 46.8 | 98.6 | 17.8/1.2 | 55.3 | 99.6 | 3 Min occluded circuit yielded rise in CO2, no significant changes |
| **Swine Surgery** | 17 | 67 | 22 | 2 | 25.3/6.4 | 41.2 | 98.3 | 20.5/1.4 | 44.4 | 94.3 | Both animals remained very stable, no adjustments needed |
| **Maximal Difference** | 17 | 67 | 32 | 2 | 33.4/13.7 | 52.7 | 97.5 | 21.3/1.4 | 42.4 | 94.5 | 12cm PEEP for swine with high PIP to maintain ventilation. This resulted in cardiac strain and instability. Sheep remained stable |

both animals could be safely coventilated at different pressures for prolonged periods [Table 3]. Statistical analysis of the composite data from the data logger demonstrates that the swine and sheep ventilation pressures were statistically different once regulated in both PIP and PEEP [Fig 4] and yet, there was no difference in the sheep's ventilation pressures throughout all testing parameters [Table 4]. Arterial blood gas analysis throughout the experiment correlated with end-tidal CO2 readings and SpO2 and confirmed the animals were maintaining adequate ventilation. Similar to the single-animal experiment, one circuit disconnect resulted in significant hypoventilation of both animals, however circuit occlusion resulted in no significant changes for the second animal, especially with pressure control ventilation. The largest ventilation pressure differential tested was 12 cmH20 between the swine and sheep (PIP/PEEP 33.4/13.7 vs. 21.3/1.4 respectively) [Fig 4]. In this scenario, both animals were being ventilated, but the high pressures ultimately caused cardiac arrythmias in the swine and the experiment was terminated.

## Discussion

A compact delivery system (capable of enabling mechanical ventilation for multiple patients from a single ventilator) potentially addresses many problems associated with acute shortages of ventilators in clinically important settings, including the current Covid-19 pandemic. This led to an initial development of split ventilation that was used in pair-matched human use including on patients during the Covid-19 pandemic [3]. However, the limited applicability of the initially described split ventilation led clinicians and scientists to the use of one-way valves, filters and flow restrictors [5].

Our work expands on this work and shows that a system can be developed to enable expanded access to life saving ventilatory support in a pressure control ventilation mode

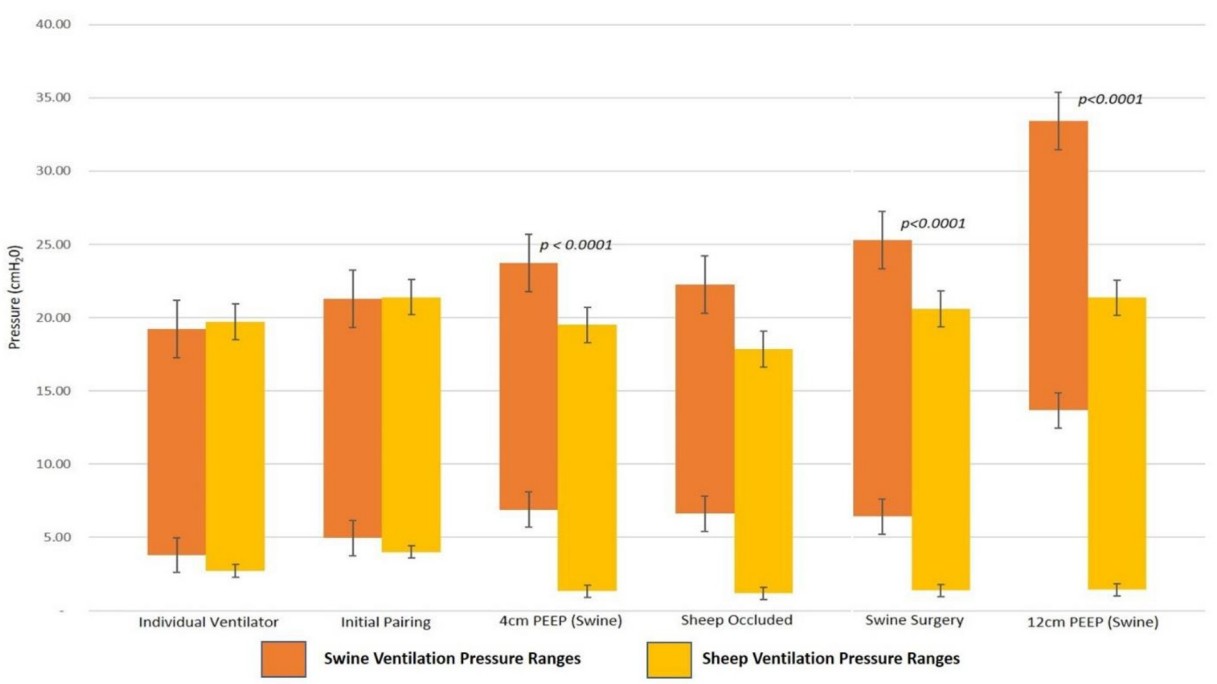

**Fig 4. Ventilation pressures across different testing conditions.** Graph shows measured average circuit pressures for the swine (orange) and sheep (yellow) during various clinical scenarios. The top of the bar represents the average PIP, while the bottom represents the average PEEP. As noted, when a PEEP booster was added to intentionally create differential pressure ventilation, the two animals were statistically different. However on individual ventilation and initial pairing, they were the same. Occlusion of the sheep's endotracheal tube did not result in any significant differences for the pig's ventilation profile.

(rather than being limited to volume control), while addressing concerns related to simple "ventilator splitting". The system developed here differs from all previous work with a *de novo* pressure regulator that was custom designed for ventilator pressures, and manufactured in an ISO compliant facility. This combined with PEEP boosters are not volume restricting and allow for differing pressures to be applied. We have demonstrated that it is possible to individualize, protect, adapt, and control ventilation in a complex, disparate dual-animal model for over 6 hours with multiple interventions [Fig 5, swine surgical procedure]. We demonstrated that one patient can experience a wide breadth of physiologic stressors and ventilatory changes, while the second patient can remain stable with unchanged parameters. This specifically addresses concerns for over/under ventilation of patients with different lung compliances. The standby mode and the rapidity with which patient can be placed on or removed from a joint circuit are especially notable.

Our system has addressed most major concerns raised regarding ventilator splitting: managing differential compliance and PEEP requirements, personalized monitoring with alarm capacity, a disconnected circuit can be simply capped if needed, and circuit occlusion does not significantly affect ventilation to the second patient.

**Table 4. Effects of different swine testing conditions on sheep ventilatory parameters.**

| | | Testing Conditions | | | | |
|---|---|---|---|---|---|---|
| Ventilatory Measurements | Individual Ventilator | Initial Pairing | 4cm PEEP (swine) | Swine Surgery | 12cm PEEP (swine) | *p* value |
| Sheep PIP (Mean ± SBD, cmH2O) | 19.72 ± 5.81 | 21.4 ± 3.63 | 19.5 ± 4.37 | 20.59 ± 0.45 | 21.36 ± 0.64 | *0.49* |
| Sheep PEEP (Mean ± SBD, cmH2O) | 2.72 ± 1.48 | 4.01 ± 0.31 | 1.34 ± 0.31 | 1.39 ± 0.19 | 1.44 ± 0.33 | *0.33* |

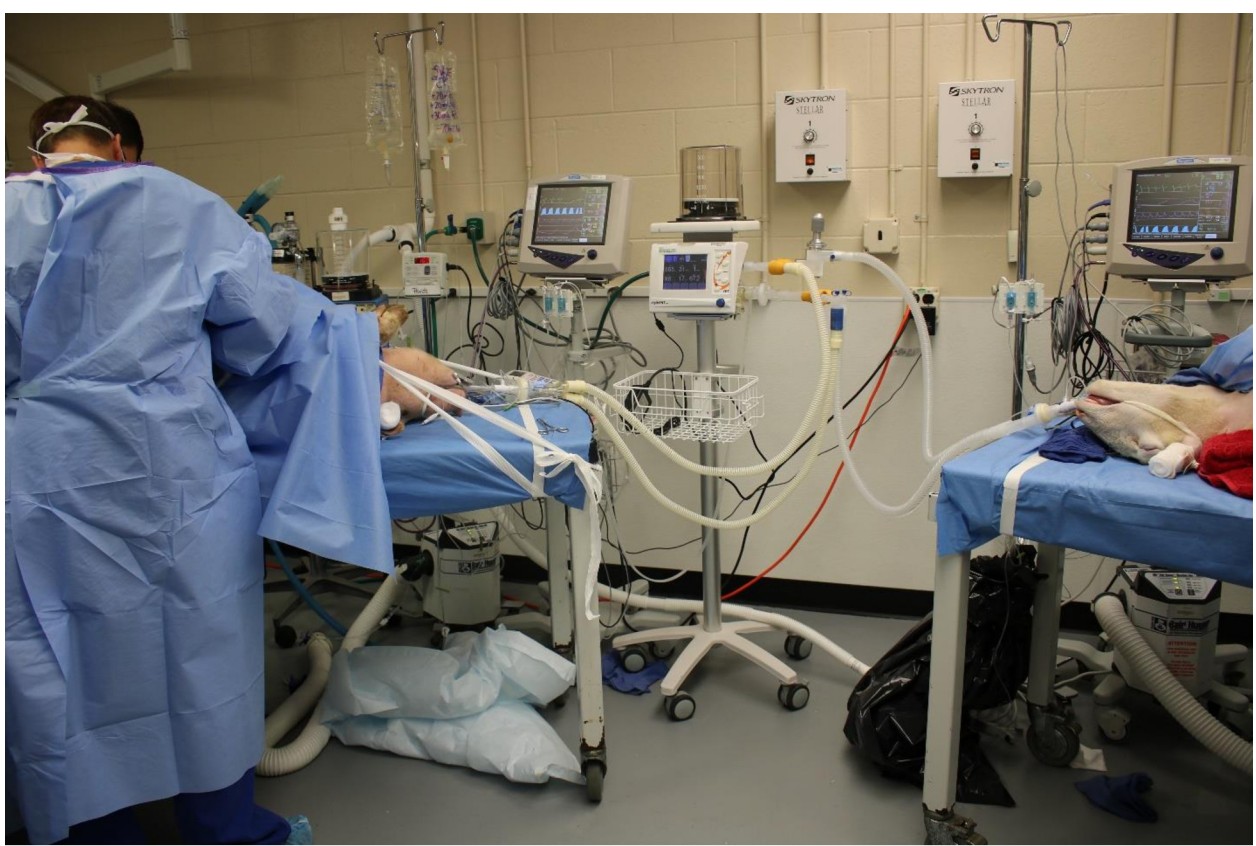

**Fig 5. Photographic demonstration of the coventilation during the swine's flap surgery.** This operation was intended to cause stress to the swine and observe for any changes in coventilation parameters for either animal. Both animals remained very stable, with unchanged ventilation parameters when compared to the prior ventilation settings.

These systems are available at a fraction of the cost, footprint and weight that would be required for comparably capable, full size ventilators. Thus, they can be deployed more rapidly than ventilators. Whether these advantages in deployability result in improved outcomes remains to be tested. This approach may make rapid and agile delivery to remote or lower-resourced locations more facile. We have shown that a system can be developed where setup and delivery time is much less than that for a full ventilator. The utilization of a standard arterial line pressure transducer and monitor allows the ability to individually monitor each patient's ventilation pressures in real time remotely. Through preassembly, systems are less subject to errors in setup than unassembled systems. It has not escaped our attention that a similar construct could be further miniaturized and made lighter, accentuating the advantages in weight and size. These potentially could even be further engineered to allow different pressures to be delivered between lungs of a single individual.

These systems do not obviate the need for medical personnel that can identify and respond to problems with the ventilator, endotracheal tube or ventilator circuit. Operators must understand ventilator mechanics and respiratory physiology. Monitoring of vital signs, oxygen saturation levels and, ideally, end-tidal $CO_2$ measurements for each patient can increase safety of any mechanical ventilator system.

There are several limitations to VentMi, including the inability to deliver variable respiratory rates or differential $FiO_2$. While potential solutions to these problems have been proposed, robust testing has not been employed. Furthermore, these results would ideally have

been confirmed with human data, but the capacity to mobilize ventilator support across the United States has limited clinical need to date. Notably, we believe split ventilation, even with the VentMi system, should *only* be utilized when all conventional ventilator resources have been exhausted. While sustained split ventilation appears feasible, we anticipate this system would be utilized only until additional ventilators could be mobilized.

In the United States, the EUA effectively allows rapid deployment of the system during the current covid19 pandemic. Other countries have similar regulatory mechanisms. Our system has emergency use authorization from the FDA for use during the Covid-19 pandemic.

## Conclusion

In settings of limited ventilator availability, delivery systems can be developed to allow increased delivery of ventilator support to enable rapid deployment under constraints of time, space and finances. The VentMi system has been efficacious in mechanical simulations and animal experiments, providing individualized pressure-control ventilation and monitoring. Optimizing ventilator support for patients where the supply of mechanical ventilators is limited will still be dependent upon ease of use, susceptibility to breakdown and the availability of medical personal.

## Supporting information

**S1 Raw data.**
(ZIP)

## Author Contributions

**Conceptualization:** Kyle K. VanKoevering, Glenn E. Green.

**Data curation:** Kyle K. VanKoevering, Anne G. Phillips, Stephen Lewis Harvey, Alvaro Rojas-Pena, Glenn E. Green.

**Formal analysis:** Kyle K. VanKoevering, Glenn E. Green.

**Funding acquisition:** Kyle K. VanKoevering, Pratyusha Yalamanchi, Glenn E. Green.

**Investigation:** Kyle K. VanKoevering, Stephen Lewis Harvey, Alvaro Rojas-Pena, Glenn E. Green.

**Methodology:** Kyle K. VanKoevering, Alvaro Rojas-Pena, Glenn E. Green.

**Project administration:** Kyle K. VanKoevering, Pratyusha Yalamanchi, Alvaro Rojas-Pena, David A. Zopf, Glenn E. Green.

**Resources:** Kyle K. VanKoevering, Pratyusha Yalamanchi, Alvaro Rojas-Pena, David A. Zopf, Glenn E. Green.

**Supervision:** Kyle K. VanKoevering, Alvaro Rojas-Pena, Glenn E. Green.

**Validation:** Kyle K. VanKoevering, Alvaro Rojas-Pena, Glenn E. Green.

**Visualization:** Kyle K. VanKoevering, Catherine T. Haring, Alvaro Rojas-Pena, Glenn E. Green.

**Writing – original draft:** Kyle K. VanKoevering, Pratyusha Yalamanchi, Catherine T. Haring, David A. Zopf, Glenn E. Green.

**Writing – review & editing:** Kyle K. VanKoevering, Pratyusha Yalamanchi, Catherine T. Haring, Anne G. Phillips, Stephen Lewis Harvey, Alvaro Rojas-Pena, David A. Zopf, Glenn E. Green.

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
