## [Decision Letter · Decision Letter 0]

11 Nov 2020

PONE-D-20-27492

Delivery system can vary ventilation parameters across multiple patients from a single source of mechanical ventilation

PLOS ONE

Dear Dr. VanKoevering,

Thank you for submitting your manuscript to PLOS ONE. After careful consideration, we feel that it has merit but does not fully meet PLOS ONE’s publication criteria as it currently stands. Therefore, we invite you to submit a revised version of the manuscript that addresses the points raised during the review process.

 Please see attached comments by the reviewers. Kindly provide point by point response in your revised manuscript

We look forward to receiving your revised manuscript.

Kind regards,

Muhammad Adrish

Academic Editor

PLOS ONE

Journal Requirements:

2. To comply with PLOS ONE submissions requirements, please provide methods of sacrifice in the Methods section of your manuscript.

3. Please clarify the relationship between the authors and the approval by the Charles River IACUC.

4.Thank you for including your ethics statement:  "University of Michigan Protocol: PRO00009358

Charles River Mattawan IACUC 1025-027".   

Please amend your current ethics statement to confirm that your named ethics committee specifically approved this study.

For additional information about PLOS ONE submissions requirements for ethics oversight of animal work, please refer to http://journals.plos.org/plosone/s/submission-guidelines#loc-animal-research  

5.Thank you for stating the following in the Competing Interests section:

[I have read the journal's policy and the authors of this manuscript have the following competing interests: KVK, GEG and DAZ are founders of MakeMedical which is the legal manufacturer of this device (provided at cost).].

Reviewers' comments:

Reviewer's Responses to Questions

**Comments to the Author**

1. Is the manuscript technically sound, and do the data support the conclusions?

Reviewer #1: Yes

Reviewer #2: Yes

2. Has the statistical analysis been performed appropriately and rigorously? 

Reviewer #1: I Don't Know

Reviewer #2: No

3. Have the authors made all data underlying the findings in their manuscript fully available?

Reviewer #1: Yes

Reviewer #2: Yes

4. Is the manuscript presented in an intelligible fashion and written in standard English?

Reviewer #1: Yes

Reviewer #2: Yes

5. Review Comments to the Author

Reviewer #1: The authors have done a remarkable work on the concept, which has potential for attenuating ventilator shortages. They have also addressed the limitation of the system which, as intriguing as it sounds, can be also very challenging especially using in high risk population.

Reviewer #2: Comments to the Author

The authors had developed the new novel delivery system to allow individualized peak inspiratory pressure settings and PEEP using a pressure regulatory valve. The system demonstrated the ability to provide ventilation across clinically relevant scenarios including circuit occlusion, unmatched physiology, and a surgical procedure, while allowing significantly different pressures to be safely delivered to each animal for individualized support. Although this paper was well written, Sufficient improvements for following issues are needed for the acceptance to the PLoS One.

Major comments

1. This individual ventilation system had improved the disadvantage and concern of previous non-individualized ventilations system. This study has clinical significance because it aims to prevent medical collapse due to the spread of COVID-19 infection.

However, I think that you should reveal how to prevent the risk of aerosol infection that may occur during suction treatment in this paper.

6. PLOS authors have the option to publish the peer review history of their article (what does this mean?). If published, this will include your full peer review and any attached files.

Reviewer #1: No

Reviewer #2: No

---

## [Author Response · Author response to Decision Letter 0]

19 Nov 2020

a. We have edited the submission to include a separate title page and reformatted the headings within the main body

2. To comply with PLOS ONE submissions requirements, please provide methods of sacrifice in the Methods section of your manuscript.

a. The methods of sacrifice were a combination of sodium pentobarbital and bilateral pneumothoraces for both animal studies. This has been added into the methods section [Line 146].

3. Please clarify the relationship between the authors and the approval by the Charles River IACUC.

a. The authors have no relationship with Charles River. Charles River is an independent animal facility which remained operational during the COVID pandemic while we were unable to complete the 2-animal study internally.

4. Thank you for including your ethics statement: "University of Michigan Protocol: PRO00009358 Charles River Mattawan IACUC 1025-027". Please amend your current ethics statement to confirm that your named ethics committee specifically approved this study. Once you have amended this/these statement(s) in the Methods section of the manuscript, please add the same text to the “Ethics Statement” field of the submission form (via “Edit Submission”).

a. Thank you, we have updated the statement as follows:

b. University of Michigan Protocol: PRO00009358

Charles River Mattawan IACUC 1025-027

The University of Michigan single-animal and Charles River dual-animal experiments were both approved by the respective animal use committees for COVID-19 Research

c. We also included the following in the methods: “All animal studies were carried out in strict compliance with the Guide for the Care and Use of Laboratory Animals of the National Institutes of Health. The protocol was reviewed by the University of Michigan University Committee on Use and Care of Animals (UCUCA) for the single pig feasibility study, and the Charles River Animal Use Committee for the combined pig and sheep study. Both studies were approved by the respective animal use committees”

5. Thank you for stating the following in the Competing Interests section: [I have read the journal's policy and the authors of this manuscript have the following competing interests: KVK, GEG and DAZ are founders of MakeMedical which is the legal manufacturer of this device (provided at cost).]. Please confirm that this does not alter your adherence to all PLOS ONE policies on sharing data and materials, by including the following statement: "This does not alter our adherence to PLOS ONE policies on sharing data and materials.” 

a. Thank you for highlighting this. Although we are the legal manufacturers, it does NOT alter our adherence to PLOS ONE policies. Please add the following into our submission:

b. “I have read the journal's policy and the authors of this manuscript have the following competing interests: KVK, GEG and DAZ are founders of MakeMedical LLC, which is the legal manufacturer of this device (provided at cost). This does not alter our adherence to PLOS ONE policies on sharing data and materials.”

6. Reviewer #1: The authors have done a remarkable work on the concept, which has potential for attenuating ventilator shortages. They have also addressed the limitation of the system which, as intriguing as it sounds, can be also very challenging especially using in high risk population

a. We thank the reviewer for their thoughtful response and appreciate their comments. We agree this system addresses many of the limitations and concerns regarding ventilator sharing, and believe it is superior to a simple splitting system. 

7. Reviewer #2: Comments to the Author

The authors had developed the new novel delivery system to allow individualized peak inspiratory pressure settings and PEEP using a pressure regulatory valve. The system demonstrated the ability to provide ventilation across clinically relevant scenarios including circuit occlusion, unmatched physiology, and a surgical procedure, while allowing significantly different pressures to be safely delivered to each animal for individualized support. 

Major comments

- This individual ventilation system had improved the disadvantage and concern of previous non-individualized ventilations system. This study has clinical significance because it aims to prevent medical collapse due to the spread of COVID-19 infection.

However, I think that you should reveal how to prevent the risk of aerosol infection that may occur during suction treatment in this paper.

a. We thank Reviewer 2 for their thoughtful comments regarding this system’s capabilities to expand ventilator support with an individualized ventilation system. With traditional vent-splitting, aerosolization is a major concern as single-room isolation is not feasible. At the endotracheal tube connection, the circuit is attached in the same fashion as traditional ventilation tubing. Thus, the system allows for a closed-line suction system to be attached to each patient’s endotracheal tube individually. These closed-line systems would dramatically reduce aerosolization as the circuit remains closed at all times, and would have no significant effect on ventilation. 

b. We have added the following to the paper to address this concern (Line 197). “Since each circuit is connected to the endotracheal tube in a similar fashion to standard ventilation, a closed-line suction system can be sued in each patient to limit aerosolization.”

---

## [Decision Letter · Decision Letter 1]

25 Nov 2020

Delivery system can vary ventilation parameters across multiple patients from a single source of mechanical ventilation

PONE-D-20-27492R1

Dear Dr. VanKoevering,

We’re pleased to inform you that your manuscript has been judged scientifically suitable for publication and will be formally accepted for publication once it meets all outstanding technical requirements.

Kind regards,

Muhammad Adrish

Academic Editor

PLOS ONE

Additional Editor Comments (optional):

Reviewers' comments:

Reviewer's Responses to Questions

**Comments to the Author**

1. If the authors have adequately addressed your comments raised in a previous round of review and you feel that this manuscript is now acceptable for publication, you may indicate that here to bypass the “Comments to the Author” section, enter your conflict of interest statement in the “Confidential to Editor” section, and submit your "Accept" recommendation.

Reviewer #2: All comments have been addressed

2. Is the manuscript technically sound, and do the data support the conclusions?

Reviewer #2: Yes

3. Has the statistical analysis been performed appropriately and rigorously? 

Reviewer #2: Yes

4. Have the authors made all data underlying the findings in their manuscript fully available?

Reviewer #2: Yes

5. Is the manuscript presented in an intelligible fashion and written in standard English?

Reviewer #2: Yes

6. Review Comments to the Author

Reviewer #2: The authors had developed the new novel delivery system to allow individualized peak

inspiratory pressure settings and PEEP using a pressure regulatory valve. Author had responded to my query.

7. PLOS authors have the option to publish the peer review history of their article (what does this mean?). If published, this will include your full peer review and any attached files.

Reviewer #2: No

---

## [Editor Report · Acceptance letter]

2 Dec 2020

PONE-D-20-27492R1 

Delivery system can vary ventilatory parameters across multiple patients from a single source of mechanical ventilation. 

Dear Dr. VanKoevering:

I'm pleased to inform you that your manuscript has been deemed suitable for publication in PLOS ONE. Congratulations! Your manuscript is now with our production department. 

Kind regards, 

on behalf of

Dr. Muhammad Adrish 

Academic Editor

PLOS ONE